# Clinical, Dermoscopic, and Molecular Features of Acantholytic Squamous Cell Carcinoma: A Systematic Review

**DOI:** 10.3390/cancers16162905

**Published:** 2024-08-21

**Authors:** Catherine Keying Zhu, Lorena Alexandra Mija, Santina Conte, Sarah Ghezelbash, Bonika Nallanathan, Geneviève Fortier-Riberdy, Margaret Redpath, Philippe Lefrançois

**Affiliations:** 1Faculty of Medicine, McGill University, Montreal, QC H3T 1E2, Canada; catherine.zhu@mail.mcgill.ca (C.K.Z.); santina.conte@mail.mcgill.ca (S.C.); 2Faculty of Medicine, Université de Montréal, Montreal, QC H3T 1J4, Canada; lorena.alexandra.mija@umontreal.ca; 3Cancer Axis, Lady Davis Institute for Medical Research, Montreal, QC H3T 1E2, Canada; sarah.ghezelbash@mail.mcgill.ca (S.G.); bonika.nallanathan@mail.mcgill.ca (B.N.); 4Division of Experimental Medicine, Department of Medicine, McGill University, Montreal, QC H3T 1E2, Canada; 5Division of Dermatology, Department of Medicine, Centre Hospitalier de l’Université de Montréal, Université de Montréal, Montreal, QC H3T 1J4, Canada; 6Department of Pathology, Jewish General Hospital, McGill University, Montreal, QC H3T 1E2, Canada; margaret.redpath@mail.mcgill.ca; 7Division of Dermatology, Department of Medicine, Jewish General Hospital, McGill University, Montreal, QC H3T 1E2, Canada; 8Department of Pharmacology & Therapeutics, McGill University, Montreal, QC H3T 1E2, Canada; 9Division of Surgical & Interventional Sciences, Department of Medicine, McGill University, Montreal, QC H3T 1E2, Canada

**Keywords:** squamous cell carcinoma (SCC), acantholytic squamous cell carcinoma (aSCC), pseudoglandular SCC, adenoid SCC, pseudovascular SCC, pseudoangiosarcomatous SCC, skin cancer

## Abstract

**Simple Summary:**

Acantholytic squamous cell carcinoma (aSCC) is a rare type of skin cancer that represents about 4.9% of all squamous cell carcinoma cases. Currently, there are no standardized guidelines for diagnosing this cancer. This study aimed to compile and analyze the existing research on the clinical and molecular characteristics of aSCC. We reviewed 52 studies that included 482 patients, most of whom were older males. We found that aSCC commonly appears on the head and neck as nodules with surface changes such as thickening, erosion, or ulceration. Our review also highlights that certain molecular markers, such as cytokeratins, are commonly present, while others, like CD15 and various vascular markers, are often absent. The findings suggest that understanding both the clinical presentation and molecular features of aSCC is crucial for accurate diagnosis. Our systematic review provides data which can help doctors better identify and treat this rare form of cancer, ultimately improving patient outcomes.

**Abstract:**

Introduction: Acantholytic squamous cell carcinoma (aSCC) is a rare clinicopathological subtype of cutaneous squamous cell carcinoma, accounting for approximately 4.9% of all SCC cases. However, there are currently no standardized criteria for the diagnosis of aSCC. This systematic review is the first to summarize the clinical and molecular features of aSCC. Methods: A systematic search of Medline, Embase, Scopus, and PubMed was performed. All articles in English or French were included, with no restriction of publication date. All articles with original data pertaining to clinical or molecular characteristics of aSCC were included. Two reviewers screened articles and resolved conflicts. Results: Our systematic review included 52 studies on the clinical and molecular features of aSCC, including a total of 482 patients (76% male, mean age at diagnosis 68.9 years): 430 cases assessed clinical features, while 149 cases assessed molecular features. The most common location of aSCC was the head and neck (*n* = 329/430; 76.5%). In terms of morphology, most lesions were described as nodules (*n* = 93/430, 21.6%), with common surface changes being hyperkeratosis (*n* = 6), erosion (*n* = 6), ulceration (*n* = 5), and crusting (*n* = 3). With regard to dermoscopy, only six cases were noted in the literature, including findings such as ulceration (*n* = 3), keratin clots (*n* = 2), and erosions (*n* = 2). Thirty-four studies discussed the molecular markers of aSCC, with the most prevalent markers being cytokeratins. CD15 negativity was noted in 23 cases, while common endothelial vascular markers such as CD34 (*n* = 16), CD31 (*n* = 15), factor VIII-related antigen (*n* = 10), and ERG (*n* = 1) were often not expressed. Finally, expression of intracellular adhesion molecules (i.e., E-cadherin, CD138) was markedly decreased compared to non-acantholytic invasive SCC. Conclusions: This systematic review summarizes the clinical characteristics and molecular features of aSCC. As clinical differentiation can be difficult, clinicopathological correlation with molecular markers may help ensure proper diagnosis.

## 1. Introduction

Acantholytic squamous cell carcinoma (aSCC), also known as pseudoglandular/adenoid (i.e., with gland-like spaces on histopathology), pseudovascular (i.e., vascular-like cords and spaces mimicking vascular differentiation), pseudoangiosarcomatous (i.e., with complex anastomosing channels lined with neoplastic cells) SCC, or adenoacanthoma of Lever, is an underreported subtype of cutaneous squamous cell carcinoma (cSCC) characterized by unique histopathological features that distinguish it from other forms of cSCC, accounting for only 4.9% of cases [1]. It is most commonly noted to occur in the sun-exposed areas of elderly individuals, with a male predominance [1]. Despite its rarity, aSCC is regarded as an intermediate- to high-risk form of SCC due to its aggressive nature and potential for metastasis, with a mortality rate ranging from 3% to 19% [2,3,4].

The diagnosis of aSCC is made histopathologically, whereby an SCC containing atypical keratinocytes with loss of cohesion between epithelial cells causes the formation of intraepithelial clefts and gives the appearance of gland-like or tubular spaces [5,6]. In some cases, these clefts appear to anastomose, giving a pseudovascular appearance. In other cases, small, rounded acantholytic cells may present, suggesting the subcategory of small-cell SCC [1]. Recent advancements in dermatopathology and molecular oncology have begun to shed light on the distinctive features of aSCC. However, these studies are often limited in scope, focusing on isolated single-center experiences or on multiple SCC subtypes rather than only developing on aSCC features [5,6].

Clinically, aSCC most commonly presents as a nodular, pink, red, or flesh-colored lesion, often associated with ulceration or crusting of the epidermis, among other presentations [7]. The macroscopic features of aSCC often represent a diagnostic challenge, as its appearance is similar to that of basal cell carcinoma (BCC) or keratoacanthoma [3]. Molecular markers associated with aSCC, such as cytokeratin (CK) and epithelial membrane antigen (EMA) markers, have been noted in some reports, yet the immunohistochemical and molecular markers specifically tied to aSCC remain under-characterized due to the scarcity of studies in the literature [1]. As a result, there remains a significant gap in comprehensive, large-scale analyses that integrate clinical, dermoscopic, and molecular data to establish standardized diagnostic criteria for aSCC. Moreover, emerging technologies, such as non-invasive optical tools and molecular biosensing, offer new avenues for diagnosis and prognosis. Still, their applicability to aSCC has not been thoroughly explored in the literature.

Given the lack of validated or standardized minimal criteria for diagnosing aSCC, this review aims to address these gaps in the current literature by compiling and synthesizing clinical, histopathological, and molecular data on the unique patterns of aSCC and their diagnostic implications. The review will emphasize the molecular landscape of aSCC, highlighting key biomarkers and their potential roles in prognosis and treatment. Additionally, it will explore the implications of these findings for future research, particularly in the development of advanced diagnostic tools and targeted therapies.

## 2. Methods

### 2.1. Search Strategy

A systematic review of the literature to assess the clinical and molecular features of aSCC was conducted following the Preferred Reporting Items for Systematic Reviews and Meta-Analyses (PRISMA) guidelines. The review was registered with PROSPERO, an international prospective registry of systematic reviews (registration CRD42024505734). A systematic search of Medline, Embase, Scopus, and PubMed was performed using the following search string: “acantholytic cutaneous squamous cell carcinoma” or “acantholytic SCC” or “acantholytic cutaneous SCC” or “pseudoglandular squamous cell carcinoma” or “pseudoglandular SCC” or “adenoid squamous cell carcinoma” or “adenoid SCC” or “adenoacanthoma of sweat glands” or “pseudovascular SCC” or “pseudovascular squamous cell carcinoma” or “pseudoangiosarcomatous SCC” or “pseudoangiosarcomatous squamous cell carcinoma” for articles published from inception to 28 January 2024.

### 2.2. Data Screening and Extraction

Title and abstract screening were conducted in duplicate by two reviewers using Covidence, online systematic review management software (www.covidence.org, accessed on 28 January 2024). No publication date restrictions were applied, and all articles in English and French were included. Articles were eligible for inclusion if they reported original data on the clinical presentations and molecular features of aSCC. Any discrepancies between reviewers (CKZ and LAM) were resolved by the senior author (PL). Data extraction was completed using a predetermined extraction form as previously established by our research group [8]. The variables examined included study design, study size, country of study, demographic features, morphological features, lesion size and location, symptomatology, dermoscopic features, and molecular findings. The quality of evidence was assessed using the Joanna Briggs Institute’s levels of evidence.

## 3. Results

Initially, 639 records were identified from all databases, and 204 original articles remained after duplicates were removed. Following full-text assessment, 52 studies were included in our review (Figure 1), with the majority being case reports (58%) (Table 1). Of the included studies, 48 evaluated clinical features, 34 studies focused on molecular features, and only 6 studies discussed dermoscopic features associated with aSCC.

### 3.1. Demographics

Our data set included 482 patients, with a male predominance (76.1%, *n* = 367).The mean age at diagnosis was 68.9 ± 14.4 years (range: 20–102). The most common ethnicity reported was Caucasian (37.6%), followed by Hispanic (11.0%) and Asian (4.6%), although a substantial number of cases (46.5%) did not report ethnicity (Table 2). Risk factors and comorbidities were only reported in 23 aSCC cases (4.8%), with the most common being history of skin cancer (*n* = 7), immunosuppression (*n* = 5), previous diagnosis of actinic keratosis (*n* = 4), and burns at the location of the lesions, including UV-induced (*n* = 4 total).

### 3.2. Clinical and Dermoscopic Features

Forty-eight studies evaluated the clinical characteristics of aSCC, and six studies further characterized dermoscopic findings in 430 patients (Table 3).

In terms of location, the majority of aSCC cases were reported on the head and neck region (*n* = 329), more precisely on the face (*n* = 36), cheek/chin (*n* = 22), eyelid (n = 18), forehead (*n* = 16), scalp (*n* = 13), ear (*n* = 14), nose (*n* = 9), lip (*n* = 8), neck (*n* = 8) and eyebrow (*n* = 7). Other reported locations included the upper limbs (*n* = 36), trunk/back (*n* = 33), lower limbs (*n* = 17), genitalia (*n* = 11) and unspecified (*n* = 4). The median tumor diameter was 3.3 cm (range: 0.2–10 cm).

A variety of morphological features were reported in aSCC. Macroscopically, aSCC lesions were most frequently described as raised nodules (*n* = 93). Other descriptors included flat-like plaque (*n* = 1), pedunculated (*n* = 1) and sessile (*n* = 1). Surface changes were commonly characterized as hyperkeratosis (*n* = 6), and erosion (*n* = 6), followed by ulceration (*n* = 5), exophytic growth (*n* = 4) and crusting (*n* = 3). Less frequently reported secondary changes were friable (*n* = 2), atrophic (*n* = 2), indurated (*n* = 2), smooth (*n* = 1), and vegetative (*n* = 1). The most common lesion color was red/erythematous (*n* = 7), followed by gray/black (*n* = 3), pink (*n* = 2), white (*n* = 2), purple (*n* = 1) and flesh-colored (*n* = 1). In terms of borders, lesions were noted to be irregular (*n* = 2) and well demarcated (*n* = 2). Lesion mobility and consistency were also reported, with mentions of firm (*n* = 2), soft (*n* = 1), fixed (*n* = 1), and mobile (*n* = 1) aSCCs. Associated symptoms reported by patients were most commonly pain (*n* = 7), pruritus (*n* = 3), purulence (*n* = 3), and hemorrhage/bleeding (*n* = 3). Only one case was noted to be painless.

Dermoscopy was only applied to aSCC lesions in six cases throughout the literature. Findings included ulcerations (*n* = 3), keratin clots (*n* = 2) and erosions (*n* = 2), with backgrounds described to be red *(n* = 3), white (*n* = 1) and pigmented (*n* = 1).

### 3.3. Molecular Features

Thirty-four studies discussed the molecular features of aSCC, with the findings summarized in Table 4. A summary of each molecular marker and its function/purpose can be found in Table 5.

Cytokeratins, keratin proteins found in the intracytoplasmic cytoskeleton of epithelial tissue, were commonly discussed in aSCC literature [9]. Most were broad-spectrum CKs/not otherwise specified (positive, *n* = 40), AE1/AE3 (positive, *n* = 20; negative, *n* = 2), CK1 (positive, *n* = 6), CK34βE12 (positive, *n* = 6), CAM5.2 (positive, *n* = 7; negative, *n* = 1), KL-1 (positive, *n* = 4), and CK5/6 (positive, *n* = 3). Other mentioned cytokeratins were CK7, CK10, CK8/18, CK19, CK20 and high-molecular-weight CK (Table 4).

Several articles discussed clusters of differentiation (CD), which are molecules expressed on the surface of leukocytes and other immune cells that can be recognized by monoclonal antibodies [10]. The most reported CD marker was CD15, which was negative in 23 cases and positive in 14 cases. Other CD markers included CD31 (negative, *n* = 15; positive, *n* = 1), CD34 (negative, *n* = 16; positive, *n* = 2), CD56 (negative, *n* = 1), CD68 (negative, *n* = 1; positive, *n* = 1), CD117 (negative, *n* = 2). Two articles discussed CD138 (syndecan 1), a marker involved in cell proliferation, whose loss of expression is related to invasion of neoplastic cells [11], and decreased expression was found in 46 cases (Table 4 and Table 5) [12,13].

Tumor suppressor genes and cell proliferation markers were also discussed. Markers that were mostly positive in aSCC cases included p63 (positive, *n* = 11), p40 (positive, *n* = 6), p21 (positive, *n* = 3), p16 (positive, *n* = 1; negative, *n* = 1) and Ki-67 (positive, *n* = 9). Negative markers were S100 (negative, *n* = 7), E6/E7 (negative, *n* = 1), desmin (negative, *n* = 1), and melan A (negative, *n* = 1) (Table 4 and Table 5).

Epithelial cell markers were also discussed, namely, EMA (CA15-3) (positive, *n* = 11, negative, *n* = 5), CEA (positive, *n* = 2; negative, *n* = 7), vimentin (positive, *n* = 16; negative, *n* = 1) and E-cadherin, whose loss of expression is associated with metastasis (positive, *n* = 2; decreased, *n* = 14). Ulex europaeus agglutinin (UEA-I), a pan-endothelial marker [14], was found to be positive in 35 cases and negative in 7 cases (Table 4 and Table 5).

Inflammatory environment markers were also noted in the aSCC literature, namely, alpha-smooth muscle actin (α-SMA), which enhances fibroblast contractile activity (positive, *n* = 5; negative, *n* = 2), and ERG [15], which typically depresses vascular inflammation (negative, *n* = 1) (Table 4 and Table 5).

Lastly, some miscellaneous markers analyzed in aSCC cases encompassed a variety of processes, such as factor VIII-related antigen, a key player in risk of thrombosis in cancer [16], which was negative in most cases (*n* = 10) and positive in a single case. Laminin-5, a marker associated with invasion of cancerous cells [17], was positive in three cases and negative in three cases. Involucrin, a protein needed for the formation of the keratinocyte layer of epithelia [18], was positive in two cases. Heat shock protein 105, which is a molecule that may be affected by malignant transformation [19], was positive in four cases and negative in two cases. Muscle-specific actin (MSA, also known as HHF35) was negative in two cases. Chromogranin A and actin/myoglobin were each negative in one case (Table 4 and Table 5).

## 4. Discussion

This systematic review comprehensively summarizes the clinical, dermoscopic, and molecular features of aSCC, highlighting its distinct characteristics and the ongoing challenges associated with its diagnosis and treatment. aSCC is a relatively rare form of SCC, accounting for less than 5% of SCC cases, and consequentially, there is very little literature available with regard to its features [5]. However, given its high risk of metastasis and poor prognosis, it is crucial that the clinical features of aSCC be recognized early and accurately, and that reliable molecular markers are available to ensure proper diagnosis. This review not only synthesizes the available evidence but also identifies critical gaps in current knowledge, particularly concerning the potential of emerging diagnostic technologies and the standardization of molecular markers for aSCC.

Our systematic review demonstrates that aSCC affects males considerably more than females (76.1% vs. 23.9%) and that aSCC is more prevalent among older adults (mean age: 68.9 years). Despite this, younger individuals can also be affected, with the youngest documented case being in a 20-year-old male [20]. These demographic trends are consistent with previous epidemiologic studies, reaffirming the need for heightened clinical awareness of aSCC in older male populations, particularly in sun-exposed areas [2].

The majority of aSCC cases being found in the head and neck region (77%), and more specifically the face, is in keeping with the existing literature [2]. In fact, most cases of aSCC are thought to be found in sun-exposed areas, which is further supported by sunburn/intentional sun exposure being risk factors for its development (Table 2). However, there were 11 cases of aSCC reported on non-sun-exposed areas, namely, on the genitalia of patients, with most cases arising on the penis [21,22,23,24,25,26,27,28,29]. Of the cases located on the genitalia, two patients were positive for human papillomavirus (HPV), while two aSCC cases arose from lichen sclerosis (LS) of the vulva. Although evidence on the association of HPV and LS is sparse, the presence of HPV and LS creates pro-inflammatory environments that can be conducive to the formation of aSCC. These findings suggest a need for further investigation into the role of HPV and LS in the pathogenesis of aSCC, particularly in non-sun-exposed areas, which could lead to more effective prevention and treatment strategies. Interestingly, two groups [30,31] demonstrated the potential effectiveness of imiquimod, an immunomodulatory drug typically used to treat HPV-induced condylomas, in treating LS, which underscores its potential role in managing aSCC associated with LS. This highlights the importance of recognizing and treating conditions such as LS and HPV infection to prevent the development of genital aSCC [30,31].

The morphological variability of aSCC poses significant diagnostic challenges. There is no widely accepted morphology for aSCC, although it has been previously reported as being nodular, flesh-colored, pink or red, and often associated with crusting or ulceration [4]. This is consistent with our findings, wherein most cases were described as nodules (*n =* 93) and having ulcerations (*n* = 5), hyperkeratosis (*n* = 6), erosions (*n* = 6) and crusting (*n* = 3). These findings reinforce the need for clinicians to maintain a high degree of suspicion for aSCC when encountering lesions with these features, especially in high-risk patients. We found the two most common colors of aSCC lesions to be red (*n* = 7) and gray/black (*n* = 3), with only one case of flesh-colored lesion. Overall, our findings suggest that aSCC most commonly presents as painful red or gray/black nodular lesions associated with crusting, hyperkeratosis, erosions and ulcerations. aSCC may clinically present very similarly to keratoacanthoma and BCC, such as when presenting with ulceration, pigmentation, rolled border and scaling [32]. However, keratoacanthoma usually has a central keratin plug creating a crateriform appearance and BCC often presents with surface telangiectasias [33,34]. Given these resemblances, it is crucial to integrate clinical, dermoscopic, and histopathological assessments to accurately differentiate aSCC from other cutaneous malignancies, thereby avoiding misdiagnosis and ensuring appropriate treatment.

Despite the growing interest in non-invasive diagnostic tools, dermoscopy has limited specificity for aSCC. Although histopathological assessment remains the gold standard for diagnosis of SCC, new non-invasive optical tools such as dermoscopy have gained widened interest. Previous research has defined indicators for well-differentiated invasive SCC (white yellowish keratin mass with hairpin vessels with white halos) and poorly differentiated SCC (erythema and irregular polymorphic vessels with ulceration and blood spots) [35]. However, dermoscopic features of aSCC were very infrequently reported in the literature, with only six studies discussing them. Some discussed dermoscopic findings were keratin clots, erosions, and ulcerations on a mostly erythematous background, although these findings are common in most SCCs and are thus not necessarily specific to aSCC. A case of aSCC also discussed dermoscopic findings of telangiectasia and ulceration, which is often seen in BCC [36]. The overlap in dermoscopic features between aSCC and other skin lesions such as BCC and keratoacanthoma underscores the need for additional research to develop more specific dermoscopic criteria or adjunctive technologies that could aid in the early and accurate detection of aSCC.

The molecular landscape of aSCC, although only partially characterized, reveals potential biomarkers that could enhance diagnostic accuracy and prognostication. There is currently no defined set of immunohistochemical or molecular markers used to investigate and characterize aSCC. The expression of cytokeratin markers such as pan-cytokeratin (AE1/AE3), broad CK, and CK34βE12 reveal the epithelial nature of the neoplasm, but would also be positive in other cutaneous SCCs [27,37]. Negative expression of common endothelial vascular markers such as ERG (*n* = 1), CD34 (*n* = 16), CD31 (*n* = 15), and factor VIII-related antigen (*n* = 10) help confirm that the tumor is epithelial in origin, which aids in ruling out angiosarcoma. However, given that staining of certain endothelial markers can be present in rare cases of aSCC or absent in angiosarcomas due to dedifferentiation, electron microscopy is crucial for its diagnostic evaluation [38]. Cases of aSCC with pseudoangiosarcomatous features have also been documented, where carcinoma tumor cells locally expressed factor VIII-related antigen (*n* = 1), which was hypothesized to be caused by the uptake of antigen-rich serum by tumor cells [38]. These findings highlight the importance of a multi-marker approach in the differential diagnosis of aSCC, particularly in distinguishing it from angiosarcomas and other malignancies with similar histopathological features. Further, our findings suggest an absence of typical melanoma markers, with the majority showing negative expression of S100 (*n* = 7) and melan A (*n* = 1), which is consistent with previously reported studies [39].

In terms of tumorigenesis and cellular proliferation, we found several cases of aSCC being positive for p63 (*n* = 11), p40 (*n* = 4) and Ki-67 (*n* = 9). However, p63 positivity is not specific to aSCC, rather it is very predictive of squamous carcinomas of epithelial origin and is highly expressed in all cSCCs with a diffuse pattern [40]. It has been previously suggested that strong nuclear Ki-67 and p63 immunostaining differentiates SCC from keratoacanthoma [41]. An article found that an average of 60–70% of aSCC tumor cells showed Ki-67 expression, suggesting a very high cell division rate [36]. High Ki-67 expression is also associated with poorly differentiated cutaneous SCC, potentially explaining aSCC’s aggressive nature in comparison to other cSCC subtypes [42]. Increased expression of p40, a marker for epithelial cells, was observed in six cases of aSCC, suggesting uncontrolled growth. The concurrent positivity for Ki-67, p40, and CK5/6 in aSCC suggests an aggressive tumor phenotype that requires careful monitoring and potentially more aggressive treatment approaches. Notably, in a review of five cutaneous aSCC patients with positive expression of Ki-67, p40, and CK from Zhan et al. [43], all cases revealed poor prognostic outcomes, including patient mortality, recurrence, and metastasis. These findings underscore the importance of these molecular markers in predicting tumor behavior and guiding clinical management. Our results support the notion that the simultaneous expression of these markers could serve as a potential biomarker for identifying aggressive and potentially invasive or metastatic aSCC, paving the way for personalized treatment strategies.

Further, although vimentin is not usually expressed in carcinomas, poorly differentiated tumors have been reported to express this intermediate filament protein [39]. A study has reported negative vimentin expression in ordinary histologic-type SCC, but positive expression in acantholytic neoplastic round cells [44]. Positive vimentin may reflect a reversal to a more embryonic, dedifferentiated phenotype, lending to a tumor’s more aggressive nature [44]. We report 16 cases of aSCC expressing vimentin, highlighting its potential as a marker in the histopathological diagnosis of aSCC and its role in identifying more aggressive forms of the disease.

Concerning histopathology, several studies on aSCC have focused on intercellular adhesion markers, revealing significant insights into the mechanisms of tumor progression. It was observed that aSCC had markedly decreased expression of E-cadherin and CD138 (46 cases of diminished syndecan 1 expression, 14 cases of diminished E-cadherin expression) compared to non-acantholytic invasive SCC [12,45]. Given the role of these markers in maintaining cellular cohesion, their loss or reduced expression is closely associated with malignant transformations observed in aSCC [12,13]. This loss of adhesion is a hallmark of the acantholysis seen in aSCC and contributes to its aggressive nature. On histology, contrary to conventional SCC, tumor cells of aSCC may show pleomorphism with hyperchromatic nuclei, irregular nuclear membranes and frequent mitotic figures, similar to those observed in high-grade malignant tumors [25]. This cellular pleomorphism, along with the distinctive acantholysis, differentiates aSCC from other SCC subtypes and underscores the importance of these markers in its diagnosis. Of note, aSCC cells are usually differentiated from keratoacanthomas by their marked cellular pleomorphism, mitotic activity and tumor cell acantholysis [41].

The diagnosis of aSCC is currently widely accepted among dermatopathologists as a histopathological one, but this reliance on histology alone presents challenges due to the variability in acantholytic features. By definition, this tumor consists of cells that have lost cohesion, creating central gland-like spaces with one to two layers of cohesive cells at the periphery, which is why it is often referred to as “acantholytic” SCC. It is also known to present with nodular, epidermal-derived proliferation that forms an appearance of lobules (or columns and island-like structures) [1]. Our systematic review highlights the importance of incorporating molecular markers (i.e., positive vimentin, decreased CD138 and E-cadherin expression) into the diagnostic criteria for aSCC. These markers, in association with clinical features (i.e., painful red or gray/black nodular lesions associated with crusting, hyperkeratosis, erosions and ulcerations), could significantly improve the accuracy of aSCC diagnosis. However, the current lack of standardized criteria for evaluating the extent of acantholysis in ambiguous cases underscores the need for stricter histopathological guidelines. Establishing more rigorous criteria may reduce the heterogeneity observed in aSCC cases and lead to better discrimination between aSCC and other cSCC subtypes.

This study has several limitations, with the most important being publication bias, as novel findings are published more than their well-defined counterparts. Further, our review is dominated by case reports (57.7%), with no randomized controlled trials (RCTs), which greatly limits the quality of evidence in the literature, making our conclusions weaker. The reliance on case reports underscores the need for higher-quality studies, such as large-scale cohort studies or RCTs, to strengthen the evidence base for aSCC. Moreover, due to the limited number of aSCC studies in the literature, especially with regard to dermoscopy, the generalizability of our results is limited. Future research should prioritize the inclusion of dermoscopic findings in aSCC case reports and series to better summarize its dermoscopic features. The specificity of our results to aSCC compared to other forms of SCC was restricted by the fact that most of the studies did not employ comparisons to other aSCC subtypes. This lack of comparative data highlights another critical gap in the literature, suggesting that future studies should aim to directly compare aSCC with other cSCC subtypes to better delineate its unique characteristics. Finally, none of the studies reported Fitzpatrick phototypes, limiting our analysis of characteristics that would be specific to certain skin tones. This omission highlights another area for improvement in future research, as understanding the influence of skin phototype on aSCC presentation could enhance diagnostic accuracy and treatment outcomes across diverse patient populations.

## 5. Conclusions

In conclusion, due to its rarity, there is a lack of consensus on the clinical and molecular findings of aSCC. Our comprehensive systematic review of the clinical characteristics and molecular features of aSCC provides a summary of these findings to facilitate its diagnosis, and highlights important gaps in our understanding of this pathology. Thus, future studies are needed to better elucidate the key features associated with aSCC on a larger scale (i.e., higher-quality studies such as RCTs), and should also focus on assessing treatment options, outcomes and prognosis of aSCC. Future research could also assess the potential use of machine learning in the diagnosis and prognosis of aSCC.

## Figures and Tables

**Figure 1 cancers-16-02905-f001:**
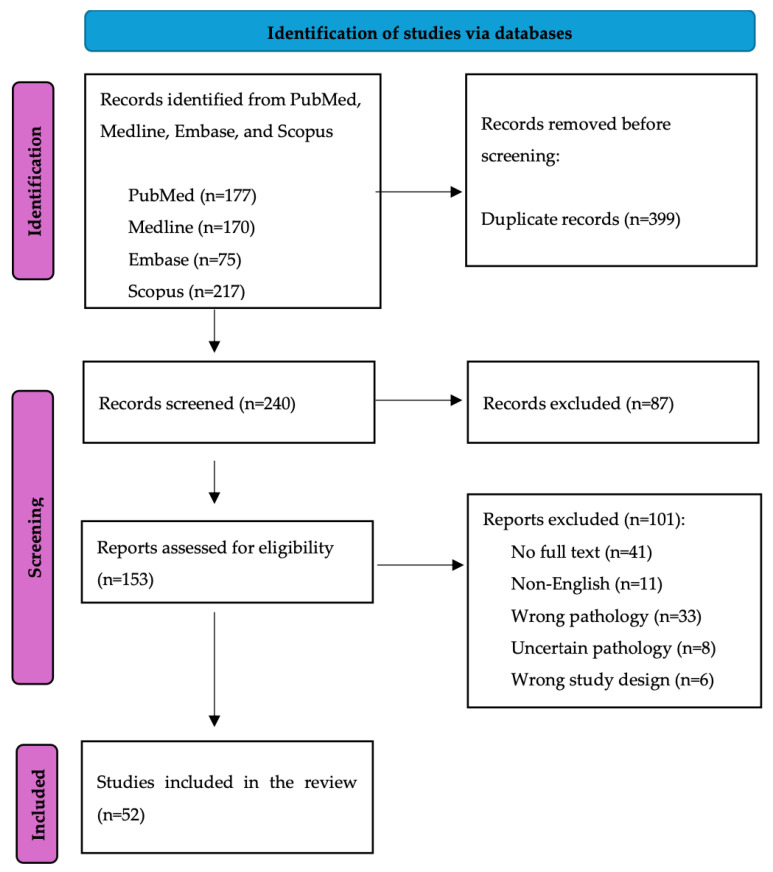
PRISMA flow diagram.

**Table 1 cancers-16-02905-t001:** Studies included in the systematic review.

Study Characteristics			Number of Studies *N* = 52, n (%)
Study type	Case reports		30 (57.7%)
	Case series		10 (19.2%)
	Prospective observational studies		3 (5.8%)
	Retrospective observational studies		9 (17.3%)
	Randomized controlled studies		0 (0%)
Quality of study	Level 1		0 (0%)
	Level 2		0 (0%)
	Level 3	3a	0 (0%)
		3b	0 (0%)
		3c	0 (0%)
		3d	4 (7.7%)
		3e	6 (11.5%)
	Level 4	4a	0 (0%)
		4b	0 (0%)
		4c	11 (21.2%)
		4d	31 (59.6%)
		4e	0 (0%)
	Level 5		0 (0%)
Country of study	United States		15 (28.8%)
	Japan		9 (17.3%)
	Spain		6 (11.5%)
	India		4 (7.7%)
	United Kingdom		3 (5.8%)
	China		3 (5.8%)
	Australia		2 (3.8%)
	South Korea		2 (3.8%)
	Other		8 (15.4%)
Study topic *	Clinical features		48 (92.3%)
	Dermoscopy features		6 (11.5%)
	Molecular features		34 (65.4%)

* Some studies may feature different topics such as clinical features and molecular features.

**Table 2 cancers-16-02905-t002:** Patient demographics.

Demographics		N (%) or Mean ± SD (Range)
Total patients		482
Age at diagnosis		68.9 ± 14.4 (20–102)
Sex (male: female), unspecified		367 (76.1%), 87 (18.0%), 28 (5.8%)
Ethnicity	Caucasian	181 (37.6%)
	Arabic	1 (0.2%)
	Asian	22 (4.6%)
	Hispanic	53 (11.0%)
	Black	1 (0.2%)
	Unspecified	224 (46.5%)
Risk factors and comorbidities	Sunburn/intentional sun exposure	2 (0.4%)
	Other burn	2 (0.4%)
	Smoking	1 (0.2%)
	Immunosuppression	5 (1.0%)
	History of skin cancer	7 (1.5%)
	History of other malignancy	4 (0.8%)
	Autoimmune condition	2 (0.4%)
	Actinic keratosis	4 (0.8%)
	Other *	11 (2.3%)
Final diagnosis	Acantholytic SCC	445 (92.3%)
	Pseudoangiosarcomatous SCC	15 (3.1%)
	Adenoid SCC	16 (3.3%)
	Pseudoglandular SCC	3 (0.6%)
	Pseudovascular SCC	3 (0.6%)

aSCC, acantholytic squamous cell carcinoma; SCC, squamous cell carcinoma; SD, standard deviation. * Other comorbidities include Bowen’s disease, epidermodysplasia verruciformis, type II diabetes, typhoid fever, nephrolithiasis, squamous sialometaplasia, herpes simplex.

**Table 3 cancers-16-02905-t003:** Short summary of clinical findings.

Clinical Features		N (%)
Number of cases		430
Location	Head and neck	329 (76.5%)
	Trunk/back	33 (7.7%)
	Upper limb	36 (8.4%)
	Lower limb	17 (4.0%)
	Genitalia	11 (2.6%)
	Unspecified	4 (0.9%)
Morphology	Flat-like plaque	1 (0.2%)
	Raised nodules	93 (21.6%)
	Erosion	6 (1.4%)
	Ulceration	54 (12.6%)
	Hyperkeratosis	6 (1.4%)
	Other	Erythematous (7) vegetative (1) exophytic (4) sessile (1) crusting (3) papillomatous (1) friable (2) smooth (1) atrophic (2) pedunculated (1) indurated (2) well demarcated (1) irregular (2) semi-mobile/mobile (1) firm (2) soft (1) fixed (1)
Symptoms	Pruritic	3 (0.7%)
	Painful	7 (1.6%)
	Painless	1 (0.2%)
	Purulent	3 (0.7%)
	Hemorrhagic/bleeding	3 (0.7%)
Size/diameter, median (range), cm		3.25 (0.2–10)
Dermoscopy		Keratin clots (2) Erosions (2) Ulcerations (3) Background: white (1), red (3), pigmented (1)
Confirmed acantholysis on histopathology		292 (67.9%)

**Table 4 cancers-16-02905-t004:** Short summary of molecular features.

Molecular Features	N (%)
Number of cases	149
Cytokeratins	AE1/AE3 positivity (20), negativity (2) CK1 positivity (6) CK5/6 positivity (3) CK7 positivity (1) CK10 positivity (1) CK8/18 negativity (1) CK19 negativity (1) CK20 negativity (1) CAM5.2 positivity (7), negativity (1) CK34βE12 positivity (6) KL-1 (pan-cytokeratin) positivity (4) High-molecular-weight CK positivity (3) Broad CK, not otherwise specified positivity (40)
Cluster of Differentiation Markers	CD15 (Leu-M1) positivity (14), negativity (23) CD31 positivity (1), negativity (15) CD34 positivity (2), negativity (16) CD56 negativity (1) CD68 positivity (1), negativity (1) CD117 negativity (2) CD138 (syndecan 1) diminished (46)
Tumor Suppressor Genes, Cell Proliferation Markers	p16 positivity (1), negativity (1) p21 positivity (3) p40 positivity (6) p63 positivity (11) E6/E7 negativity (1) S100 negativity (7) Desmin negativity (1) Ki-67 positivity (9) Melan A negativity (1)
Epithelial Tissue Markers	EMA positivity (11), negativity (5) UEA-I positivity (35), negativity (7) Vimentin positivity (16), negativity (1) E-cadherin positivity (2), decreased (14) CEA positivity (2), negativity (7)
Markers for Inflammatory Environments	a-SMA positivity (5), negativity (2) ERG negativity (1)
Miscellaneous	Factor VIII-related antigen positivity (1), negativity (10) Laminin-5 positivity (3), negativity (3) MSA/HHF35 negativity (2) Involucrin positivity (2) HSP105 positivity (4), negativity (2) Chromogranin A negativity (1) Actin/myoglobin negativity (1)

CAM, cell adhesion molecule; EMA, epithelial membrane antigen; ERG, ETS-related gene; a-SMA, a-smooth muscle actin; HHF35, muscle-actin specific monoclonal antibody; HSP105, heat shock protein 105; MSA, mammary serum antigen; UEA, ulex europaeus agglutinin.

**Table 5 cancers-16-02905-t005:** Markers positively or negatively associated with aSCC, with corresponding dermatopathological purpose. Information taken from https://www.pathologyoutlines.com/ (accessed on 24 March 2024).

Marker	Function
AE1/3	Broad-spectrum cytokeratin marker for cytokeratins 1–8, 10, 14–16 and 19. Immunoreactivity observed in epithelia and most carcinomas (i.e., tumors of epithelial origin).
a-SMA	There are three isoforms of smooth muscle actin: alpha, beta and gamma. Alpha actins are found in muscle tissues and required for contraction. Usually negative in squamous cell carcinomas.
CAM5.2	Usually positive in glandular epithelia and adenocarcinomas, whereas negative in squamous epithelium and squamous cell carcinomas.
CA15-3 (EMA, MUC1)	Epithelial membrane antigen highly expressed by most carcinomas (e.g., adenocarcinomas, squamous cell carcinomas) and hematologic cancers.
CD15	Mostly used for diagnosis of Hodgkin’s lymphoma. May be a granulocyte marker.
CD31	Most sensitive and specific endothelial marker in paraffin sections. Negative in adenomatoid tumor and pseudoangiomatous stromal hyperplasia of the breast.
CD34	Distinguish Kaposi sarcoma, dermatofibrosarcoma protuberans, and epithelioid sarcoma from dermatofibroma; distinguish solitary fibrous tumor from desmoplastic mesothelioma; distinguish hemangiopericytoma from endometrial stromal sarcoma.
CD56	Marker of natural killer (NK) cells and NK lymphomas.
CD68	Lysosomal marker used to identify histiocytic and monocytic cells. Usually used to diagnose histiocytic sarcoma.
CD117	Proto-oncogene activated in gastrointestinal stromal tumors (GISTs). Negativity is associated with leiomyoma, leiomyosarcoma, smooth muscle tumors and solitary fibrous tumors.
CD138 (syndecan 1)	Involved in cell proliferation, migration, adhesion and angiogenesis, with loss of CD138 expression leading to enhanced motility and invasion of neoplastic cells. Low or absent expression in testicular germ cell tumors, sarcomas and melanomas.
CEA	Also known as carcinoembryonic antigen. Usually considered epithelial marker with expression in many adenocarcinomas, lung squamous cell carcinoma and sweat gland carcinomas. Negative staining in melanomas, acantholytic squamous cell carcinoma (pseudoglandular).
Chromogranin	Commonly used neuroendocrine marker. Specific for neuroendocrine cells, but not sensitive.
CK34βE12	Positive staining in classic and basaloid squamous cell carcinoma, as well as amyloid deposits associated with squamous cell carcinoma and dysplasia in the head and neck.
CK1	Highest-molecular-weight keratin. Positive staining associated with keratinizing squamous cell carcinoma.
CK5/6	Together with p63, used to detect squamous cell origin in poorly differentiated carcinomas.
CK7	Generally negative (with some variation) in colorectal carcinoma, Merkel cell carcinoma, hepatocellular carcinoma, prostatic adenocarcinoma, adrenocortical tumors and squamous cell carcinoma.
CK10	Defects in CK10–CK1 protein network cause structural instability and weakness of keratinocytes, causing blistering, hyperproliferation and hyperkeratosis.
CK8/18	Positive in bile duct, invasive ductal breast, hepatocellular, neuroendocrine, pancreatic, prostatic, renal cell, and squamous cell (cervical and oral) carcinomas. Usually negative in smooth muscle tumors.
CK19	Present in simple and complex epithelium; positive staining in hair follicles; negative stain in trichilemmoma.
CK20	Epithelial marker; positive staining in Merkel cell carcinoma and fibroepithelioma of Pinkus. Usually negative staining in squamous cell carcinomas.
High-molecular-weight CK	High-molecular-weight cytokeratins 1–6, 10, 14, 15 and 16.
Desmin	Good screening marker for neoplasms with myogenic differentiation such as rhabdomyosarcoma, rhabdomyoma, leiomyosarcoma, leiomyoma and smooth muscle.
E-cadherin	Transmembrane protein involved in cellular adhesion where loss is associated with tumor progression, chemoresistance and metastases.
ERG	Marker of endothelial differentiation including vascular neoplasms. Expression usually negative in all non-prostatic carcinomas.
Factor VIII-related antigen	Common endothelial marker with positive staining of endothelial cells, megakaryocytes, platelets and mast cells.
HSP105 ^1^	Heat shock proteins are a group of heat stress proteins that may be affected by malignant transformation. Expression is decreased in cutaneous squamous cell carcinomas, but poorly differentiated subtypes show higher expression.
Involucrin	Contributes to formation of the insoluble cell envelope with loricrin.
Ki-67	Marker of cell proliferation. Commonly increased in most malignant and inflammatory conditions.
KL-1 (pan-cytokeratin)	Broad-spectrum keratin antibody for CK1–4, 10–11 or CK1, 2, 5–8, 11, 14, 16–18. Present in most carcinomas.
Laminin-5 ^2^	Associated with invading cancer cells such as cervical carcinomas and cutaneous squamous cell carcinoma invasion.
Melan A	Melanocyte lineage-specific marker in the diagnosis of metastatic melanoma. Also positive in nonmelanocytic tumors with melanosomes (e.g., angiomyolipoma, PEComa, lymphangioleiomyomatosis). Negative in most carcinomas, lymphomas.
MSA (HHF35)	Identifies skeletal muscle, smooth muscle cells. Known for negative staining in angiomyofibroblastoma and epithelioid mesothelioma.
p16	Tumor suppressor protein that prevents progression into S phase of cell cycle. Protein function is silenced in many HPV and non-HPV (e.g., breast, colon, pancreatic, head and neck, melanoma)-related tumors.
p21	Negative cell cycle regular in G2–M phase and G1–S phase.
p40	Stimulates cell proliferation and favors unrestrained tumor growth. Nuclear marker with expression in squamous, urothelial, myoepithelial cell carcinomas.
P63	Important regulator of epidermal keratinocyte proliferation and embryonic epidermal growth. Usually positive in squamous and basal cell carcinomas and helps differentiate from melanomas.
S100	Marker of melanocytes and useful for evaluating nerve sheath tumors and melanoma.
UEA-1 ^3,4^	Ulex europaeus agglutinin I is a plant lectin with an affinity for L-fucosyl residues associated with dorsal root ganglion neurons and their axonal processes. In the context of skin carcinomas, UEA-1 stains positive in endothelial cells and negative/weakly positive in the epidermis.
Vimentin	Intermediate filament for mesenchymal tissues. Usually positive in melanomas, negative for carcinomas (but many exceptions) and epithelial tumors.

^1^ https://www.ncbi.nlm.nih.gov/pmc/articles/PMC8213956/ (accessed on 19 March 2024). ^2^ https://www.nature.com/articles/bjc2011283 (accessed on 19 March 2024). ^3^ https://pubmed.ncbi.nlm.nih.gov/2479197/ (accessed on 19 March 2024). ^4^ https://pubmed-ncbi-nlm-nih-gov.proxy3.library.mcgill.ca/8708150/ (accessed on March 19 2024).

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
