# Peer review of "Clinical, Dermoscopic, and Molecular Features of Acantholytic Squamous Cell Carcinoma: A Systematic Review"

_cancers, 2024, doi:10.3390/cancers16162905_

Round 1

Reviewer 1 Report

Comments and Suggestions for Authors

The manuscript ID cancers-3148923 mainly presents an analysis about the progress in particular clinical, dermoscopic and molecular features exhibited by acantholytic squamous cell carcinoma. Please see below a list of comments to the authors:

  1. The authors should clearly state in the abstract their originality and what this report adds to literature in respect to previous publications in the topic.
  2. Advantages in the study of molecular findings of aSCC should be better described in the introduction section.
  3. A graphical abstract including the motivation and the main findings would be welcome to easily visualize the importance of the work.
  4. A roadmap for describing the evolution of the research in these clinical, dermoscopic and molecular features studied would improve the potential impact of the work.
  5. The progress in the different techniques employed in clinical, dermoscopic and molecular features of aSSCC should be better described.
  6. How is currently the error bar in the clinical differentiation of aSCC?
  7. Please describe better details about the potential of clinico-pathological correlation with molecular markers.
  8. The authors are invited to describe the attractive assistance of mathematical functions, machine learning tools or nanoscale effects for this study. You can see comparative topics in biosensing as: https://doi.org/10.3390/s19214728
  9. More perspectives should be included in the discussion section. You can see for instance: https://doi.org/10.3390/cancers16030664
  10. In my opinion, the citations presented in collective form should be split in order to better justify the individual importance of each reference selected for this systematic review.

Comments on the Quality of English Language

A proofreading is suggested

Author Response

Thank you and the reviewers for their constructive comments and the opportunity to revise our manuscript. We have made changes to the manuscript based on these comments and believe the paper has been significantly strengthened as outlined in our point-by-point response below. Our changes in the manuscript are tracked for your convenience. The line numbers in our point-by-point response refer to the manuscript with tracked changes.

Comments

Response

Reviewer 1 comments

1. The authors should clearly state in the abstract their originality and what this report adds to literature in respect to previous publications in the topic.

1. Thank you very much for the suggestion. We have added in the introduction of our abstract describing the importance and novelty of our work (lines 4-6 of p.3)

2. Advantages in the study of molecular findings of aSCC should be better described in the introduction section.

A graphical abstract including the motivation and the main findings would be welcome to easily visualize the importance of the work.

A roadmap for describing the evolution of the research in these clinical, dermoscopic and molecular features studied would improve the potential impact of the work.

The progress in the different techniques employed in clinical, dermoscopic and molecular features of aSSCC should be better described.

2. Thank you for this remark, we have now added emphasis to the role of molecular markers in aSCC (p. 6 lines 3-5)

The suggestion of a graphical abstract is indeed an interesting idea. We have now provided a possible graphical abstract in p.4. Please let us know if you would like to bring additional changes to it.

Indeed, we agree that a roadmap to describe the evolution of research and techniques employed in aSCC could be interesting to include. However, due to the limited number of studies, we believe it is difficult to report of this.

3. How is currently the error bar in the clinical differentiation of aSCC?

Thank you for your question. Unfortunately, due to the rarity of this disease, there aren’t studies to date commenting on the error bar in the clinical differentiation of aSCC.

4. Please describe better details about the potential of clinico-pathological correlation with molecular markers.

Thank you for the following suggestion. We believe that we have described in depth the pathological correlation of molecular markers with histological findings in our discussions section (p.13-16). If you would like further correlations, please provide us with an example of clinico-pathological correlation you would like us to comment on.

5. The authors are invited to describe the attractive assistance of mathematical functions, machine learning tools or nanoscale effects for this study. You can see comparative topics in biosensing as: https://doi.org/10.3390/s19214728

More perspectives should be included in the discussion section. You can see for instance: https://doi.org/10.3390/cancers16030664

Thank you for this idea. However, after screening the literature, we haven’t found articles discussing the use of mathematical functions, machine to assist in the diagnosis of aSCC.

We have however added a sentence in our conclusion to call for future studies to assess the potential use of machine learning in this topic (p.18 lines 1-2)

6. In my opinion, the citations presented in collective form should be split in order to better justify the individual importance of each reference selected for this systematic review.

The references are listed in order of appearance in the manuscript as standard. Please provide us with an example of how we should split of references if you believe it is a necessity.

Reviewer 2 Report

Comments and Suggestions for Authors

This systematic review of acantholytic squamous cell carcinoma is supported by 52 articles, considering the clinical, molecular, and dermoscopic features of 482 patients. To date, the greatest number of cases have been reported. The most important findings were the predominance of cytokeratins and tumor suppressor genes in the head and neck areas. This calls for further high-quality and diversified research on aSCC.

Intoroduction:

1. Brief definitions for certain terms such as "pseudoangiosarcomatous" may help facilitate understanding.

2. Thus, the introduction is implicit. A clearer introduction that highlights the importance of aSCC, the challenges, and what the paper is trying to achieve will provide a better context.

3. The use should be standardized to just one term for aSCC and, where appropriate, provide brief definitions for truly complex terms.

4. They either provide a narrow range of mortality or explain the reasons for such variability.

5. Some of the fragmented sentences may be merged to improve readability as follows: "In some cases, these clefts appear to anastomose, giving a pseudovascular appearance. In other cases, small rounded acantholytic cells may present, suggesting the subcategory of small cell SCC."

Methods

6. This section lacks an introduction that puts in context why a systematic review is being conducted. One or two sentences explaining its importance and the goals of the review will help set up the role.

7. Adding methodological details about the steps during data extraction and how conflicts are handled adds to transparency and reproducibility.

8. PROSPERO must be briefly explained along with its importance in systematic reviews. This can be either in a footnote or as a parenthesis.

Results and discussion

9. Being intrinsically understandable, the study type was dominated by case reports (57.7 %). This, in itself, is a limitation, as case reports provide weaker evidence than RCTs or observational studies. Further, the fact that there are no RCTs at all means that there is a gap in the highest levels of presently existing evidence for aSCC and needs to be strongly pointed out as a very important area for future research.

10. The need for higher-quality studies, particularly RCTs, to strengthen the evidence base for aSCC cannot be overemphasized.

Author Response

Thank you and the reviewers for their constructive comments and the opportunity to revise our manuscript. We have made changes to the manuscript based on these comments and believe the paper has been significantly strengthened as outlined in our point-by-point response below. Our changes in the manuscript are tracked for your convenience. The line numbers in our point-by-point response refer to the manuscript with tracked changes.

Reviewer 2 comments

Introduction:

1. Brief definitions for certain terms such as "pseudoangiosarcomatous" may help facilitate understanding.

2. Thus, the introduction is implicit. A clearer introduction that highlights the importance of aSCC, the challenges, and what the paper is trying to achieve will provide a better context.

3. The use should be standardized to just one term for aSCC and, where appropriate, provide brief definitions for truly complex terms.

4. They either provide a narrow range of mortality or explain the reasons for such variability.

5. Some of the fragmented sentences may be merged to improve readability as follows: "In some cases, these clefts appear to anastomose, giving a pseudovascular appearance. In other cases, small rounded acantholytic cells may present, suggesting the subcategory of small cell SCC."

1. Many thanks for taking the time to review our paper. As per your suggestion, we have now included short descriptions of the following terms “pseudoglandular”, “pseudovascular”, “pseudoangiosarcomatous” (p.5 lines 4-7)

2. Thank you for this suggestion. We believe that our introduction already highlights the importance of aSCC, as it can have a relatively high mortality rate (p.5 lines 11-12). Our paper tries to create a comprehensive summary of the clinical and molecular features of aSCC given the lack of validated diagnostic criteria (p.6 lines 7-10)

3. In our paper, we refer to acantholytic squamous cell carcinoma (aSCC) using only one term. In the first paragraph of the introduction, we are stating other possible terms that have been used in previous literature for aSCC. We have provided definitions for the other terms (p.5 lines 3-6)

4. The larger range of mortality is because there exist very few cohort studies assessing the mortality in aSCC. For instance one study of 155 patients, reports a mortality rate of 3% vs a study of 35 patients reports a rate of 19%. The references in question are seen in the manuscript (p.5 lines 11-13).

5. Thank you very much for this suggestion to improve the flow of our paper. We have adjusted the sentence in question (p.5 lines 18-19)

Methods

6. This section lacks an introduction that puts in context why a systematic review is being conducted. One or two sentences explaining its importance and the goals of the review will help set up the role.

7. Adding methodological details about the steps during data extraction and how conflicts are handled adds to transparency and reproducibility.

8. PROSPERO must be briefly explained along with its importance in systematic reviews. This can be either in a footnote or as a parenthesis.

6. Thank you for pointing this out. We had stated the objective of our study in the last paragraph of our introduction. However, for clarity, we have added this objective in the first sentence of our methods (p.6 line 11)

7. The methodology of our study pertaining to data extraction and conflict resolution are explained in p.7 lines 9-10. We have further provided as reference another study performed by our group where a similar extraction method was used.

8. We have now explained what PROSPERO is and its importance in systematic reviews (p.6 line 16-17)

Results and Discussion

9. Being intrinsically understandable, the study type was dominated by case reports (57.7 %). This, in itself, is a limitation, as case reports provide weaker evidence than RCTs or observational studies. Further, the fact that there are no RCTs at all means that there is a gap in the highest levels of presently existing evidence for aSCC and needs to be strongly pointed out as a very important area for future research.

10. The need for higher-quality studies, particularly RCTs, to strengthen the evidence base for aSCC cannot be overemphasized.

9, 10) Thank you very much for bringing up this limitation. We have now commented on the lack of RCTs in the literature of aSCC and emphasized the need for higher-quality studies in the future (p.17 lines 5-8).

Round 2

Reviewer 1 Report

Comments and Suggestions for Authors

The authors have clarified some points raised in the initial review stage; however,

fundamental issues to improve the presentation of the work are still present, please see

below two examples previously pointed out:

1. From the text it is not clear the cutting-edge progress and the starting point of this

systematic review regarding that a roadmap is missing as suggested.

2. The perspectives supported by previous analysis in the topic of biosensing are very

limited to easily visualize future work with this paper as a base for future research.

Comments on the Quality of English Language

A proofreading is suggested 

Author Response

Comments 1: 1. From the text it is not clear the cutting-edge progress and the starting point of this systematic review regarding that a roadmap is missing as suggested.

Response 1: 

  1. Thank you very much for the suggestion and for your revision of our manuscript. We have revised the introduction and underlined more clearly the cutting-edge progress in the field and the starting point of our systematic review. Additionally, we have included a roadmap to better introduce the structure and content of this review. We hope these further clarify the context and objectives of this manuscript. This roadmap of the systematic review is summarized in p.6 lines 12-18.

Comments 2: 2. The perspectives supported by previous analysis in the topic of biosensing are very limited to easily visualize future work with this paper as a base for future research.

Responses 2: 

  1. Thank you for this remark. We have expanded the discussion to better articulate how our findings can serve as a foundation for future research, particularly in regard to biosensing and the development of diagnostic tools, as you suggested. We hope this clarifies the future implications of our work and how it could guide subsequent studies in this area.

The following improvements can be found at

- p.12 lines 5-7 and 12-14

- p.13 lines 2-4 and 14-16

- p.14 lines 15-18

- p.14 lines 9-10 and 17-18

- p.17-18 lines 23-3

  • p.18 lines 8-10, 12-13, 15-17 and 19-21

Comments 3: Comment on Quality of English language: A proofreading is suggested

Responses 3: Thank you for your suggestions. We submitted the text for an additional language review and now believe that the English level has been improved and is ready for publication.

Reviewer 2 Report

Comments and Suggestions for Authors

No further comments. Thank you!

Author Response

Comments 1: 

  1. No further comments. Thank you!

Responses 1: 

  1. Thank you very much for your review of our manuscript!

Round 3

Reviewer 1 Report

Comments and Suggestions for Authors

The authors have improved the presentation of their work, and in my opinion, the analysis can be useful for future research. Then, I can recommend this paper for publication as it is.

Comments on the Quality of English Language

A proofreading is suggested. Specially see figure 1.